# Early antiretroviral therapy and its impact on natural killer cell dynamics in HIV-1 infected men who have sex with men: a cross-sectional pilot study evaluating the impact of early ART initiation on NK cell perturbation in HIV infection

Matrona Akiso,[1,2] Daniel Muema,[1,3] Robert Langat,[1,4] Kewreshini K. Naidoo,[3] Geoffrey Oino,[1] Gaudensia Mutua,[5] Christina Thobakgale,[6,7] Daniel Ochiel,[5] Kundai Chinyenze,[5] Omu Anzala,[1,2] Marianne W. Mureithi[1,2]

**ABSTRACT**   Phenotypic changes and functional impairment of natural killer (NK) cells occur early in HIV-1 infection. Antiretroviral therapy (ART) effectively restores CD4+ T cell counts and suppresses HIV-1 to undetectable levels. The role and efficacy of immediate ART initiation in mitigating NK cell aberrations remain to be elucidated comprehensively. This study hypothesized that HIV-1 infection negatively influences NK cell evolution and that early ART initiation restores these perturbations. Blood samples were collected longitudinally from five acutely HIV-1 infected men who have sex with men in Nairobi, Kenya. Participants were immediately initiated on ART after HIV-1 diagnosis. Blood samples were drawn pre-infection and at sequential bi-weekly post-infection time points. Peripheral blood mononuclear cells were stained with panel NK cells surface markers to assess HIV-induced phenotypic changes by flow cytometry. Some cells were also stimulated overnight with K562 cell line, IL-2, and IL-15 and stained for flow cytometry functionality. HIV-1 infection was associated with significant reductions in the production of IFN-γ ($P = 0.0264$), expression of CD69 ($P = 0.0110$), and expression of NK cell inhibitory receptor Siglec7 ($P = 0.0418$). We observed an increased NK cell degranulation ($P = 0.0100$) and an upregulated expression of cell exhaustion marker PD-1 ($P = 0.0513$) at post-infection time points. These changes mainly were restored upon immediate initiation of ART, except for Siglec7 expression, whose reduced expression persisted despite ART. Some HIV-associated changes in NK cells may persist despite the immediate initiation of ART in acute HIV-1 infections. Our findings suggest that understanding NK cell dynamics and their restoration after ART can offer insights into optimizing HIV-1 treatment and potentially slowing disease progression.

**IMPORTANCE**   Natural killer (NK) cells play a crucial role in controlling of HIV-1 replication and progression to disease. Perturbations of their functionality may therefore result in deleterious disease outcomes. Previous studies have demonstrated reduced NK cell functionality in chronic HIV-1 infection that positively correlated to HIV-1 viral load. This may suggest that control of HIV-1 viremia in acute HIV-1 infection may aid in enhancing NK cell response boosting the inate immunity hence effective control of viral spread and establishment of viral reservoir. Antiretroviral therapy (ART) effectively supresses HIV-1 viremia to undectable levels and restores CD4+ T cell counts. Our study highlights the significant role of early ART initiation in mitigating NK cell disruptions caused by acute HIV-1 infection. Our results suggest that early initiation of ART could have benefits beyond suppressing viral load and restoring CD4+ T cell counts. In addition, it could boost the innate immunity necessary to control disease progression.

Address correspondence to Marianne W. Mureithi, marianne@uonbi.ac.ke.

The authors declare no conflict of interest.

See the funding table on p. 5.

This work was presented as an oral presentation on the 24th of January, 2023 at the UoM/UoN STD/HIV/SRH COLLABORATIVE RESEARCH GROUP ANNUAL REVIEW MEETING 2023 held in Nairobi, Kenya. The work was also presented as a poster at the 18th International Congress of Immunology held from 27th November to 2nd December 2023 in Cape Town, South Africa

**KEYWORDS** acute HIV-1 infection, ART, innate immunity, natural killer cells; MSMs, seroconversion

The advent of antiretroviral therapy (ART) has revolutionized HIV management by restoring CD4+ T cell counts and mitigating HIV viral loads (1). Beyond impacting CD4+ T cells, HIV infection substantially influences the phenotype and functionality of innate immune cells like natural killer (NK) cells (2, 3). Several studies have correlated a decrease in NK cell functionality during HIV infection with HIV disease progression (1, 4). However, there are discrepant results from different studies regarding the restoration of NK cell functionality after successful ART. Given the seminal role of NK cells in innate immunity and their pivotal position as adaptive immunity modulators, this pilot study sought to longitudinally assess the impact of early initiation of ART on the restoration of phenotypic development and effector functions of NK cells in early HIV-1 infection.

Five acutely HIV-1 infected men who have sex with men, aged 18 years and above, and participating in the International AIDS Vaccine Initiative's protocol Simulated Vaccine Efficacy Trial (SiVET: P137/03/2015) in Nairobi, Kenya, were enrolled in the study. Refer to Table S1 and Fig. S1a for the participants' demographic data and the series of events at different time points pre and post ART initiation. The study was approved by the Ethics and Review Committee of Kenyatta National Hospital, University of Nairobi (P61/02/2017). Participants were immediately initiated on ART after HIV-1 diagnosis. Blood samples were drawn pre-infection and at sequential bi-weekly post-infection time points. Peripheral blood mononuclear cells (PBMCs) were isolated as previously described (5) and stained with a panel of NK cell surface markers to assess HIV-induced phenotypic changes by flow cytometry (activation panel). Some cells (functional panel) were also stimulated overnight with K562 cell line, IL-2, and IL-15 and stained to assess NK cell functionality (flow cytometry antibody panels are shown in Table S2). Gating and definition for the different NK cell subsets was as illustrated in Fig. S2. Statistical analysis was performed using GraphPad Prism software (version 8.0.1). Comparison between time points was done using a two-tailed paired t test while comparisons among more than two time points were done using unmatched, non-parametric one-way analysis of variance (ANOVA) with multiple comparisons. The level of significance was calculated at 95% confidence interval ($P \leq 0.05$).

First, we assessed the effect of early initiation of ART on NK cell populations, absolute CD4 counts, and HIV-1 viral loads. We observed no changes in total NK cell frequencies and absolute CD4 counts across different time points. As expected, the HIV-1 viral loads (copies/mL) decreased over time after ART initiation (Fig. S2b through d). From the activation panel, we measured the expression of cell activation markers CD38, CD69, and HLA-DR. Compared to the pre-infection time point, there was no change in the expression of CD38 and HLA-DR but there was a reduction in the expression of CD69 that was restored about 30 days upon ART initiation (Fig. 1a through c). We also evaluated the expression of the cell exhaustion marker PD-1 in the total NK cell population and noted its increased expression from the pre-infection time point to 0–5 days after ART (Fig. 1d; $P = 0.0513$) but decreased to levels beyond the pre-infection time point by 11–15 days after ART (Fig. 1d; $P = 0.0140$). We observed a reduction in the expression of inhibitory marker Siglec7 in overnight stimulated NK cells by the 0–5 days after ART compared to the pre-infection time point (Fig. 1f; $P = 0.0093$), and this was sustained to 25–30 days after ART. Changes in the frequency of cells expressing the phenotypic markers are shown is Fig. S3.

We evaluated the functionality of the total NK cell population and the NK cell subsets after HIV-1 infection. In both, there was a significant increase in the expression of CD107a from the pre-infection time point to about 2 weeks after ART initiation (Fig. 2a through e) that was restored around 1 month after ART initiation. Contrary to the degranulation, production of IFN-γ by NK cells was reduced early in HIV-1 infection. This decrease was restored in both the total NK cell population and in the NK cell sub populations apart from the CD56bright and the CD56dim naïve NK cell population 1 month after ART. Similar

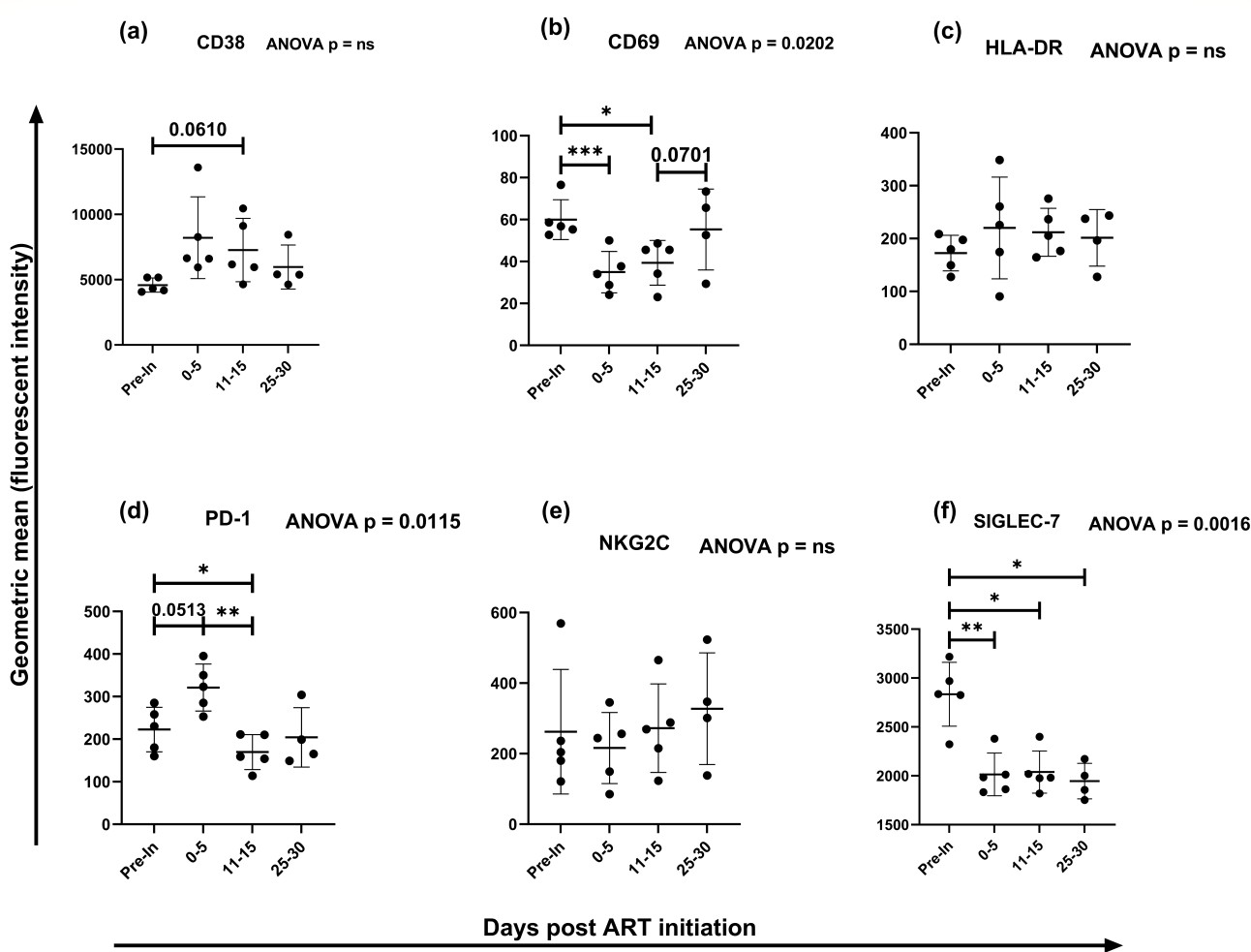

**FIG 1** Expression levels of NK cell phenotypic markers. Cell activation markers CD38 (a), CD69 (b), and HLA-DR (c); cell exhaustion marker PD-1 (d); NK cell activation marker NKG2C (e); and NK cell inhibitory maker Siglec7 (f) on total NK cell population in pre-infection (pre-in) and in early HIV-1 infection at different time points upon ART initiation. Statistical test used between time points: two-tailed paired t test. Statistical test used across time points: unmatched, non-parametric one-way ANOVA with multiple comparisons [mean of each column with the mean of a control column (pre-infection time point); Geisser-Greenhouse correction used]. Each symbol indicates a participant. Line indicates mean value. Error bars indicate standard deviation. *$P < 0.05$, **$P < 0.01$, ***$P < 0.001$.

trends were observed in the frequency of cells producing IFN-γ. Detailed illustrations for the NK cells functional changes are shown in Fig. 2 and Fig. S4.

Our study demonstrated early perturbations of NK cell phenotypes and functionality following HIV-1 infection. We observed significant reductions in the intensity and frequency of total NK cells expressing CD69, a marker for early cell activation. These findings are congruent to others reported elsewhere (6) that noted significant reductions of CD69 expression levels by NK cells and the NK cells expressing CD69 in in HIV-1 infected individuals compared to healthy individuals (6). Our prospective study revealed a restoration of CD69 levels to near pre-infection levels 1 month after ART initiation. It will be interesting to unravel the mechanism by which HIV-1 leads to this downregulation as this could be a possible therapeutic target. Increased levels of PD-1 were associated with limited NK cell proliferation in a study by Norris et al. (7). Therefore, both our results and those of Norris' group suggest that HIV-1 infection could possibly interfere with NK cell proliferation as a way to evade the body's first line of immunity.

A previous study showed that Siglec7 receptor defines highly functional NK cells (8). In our study, we observed a reduced production of IFN-γ and this could be explained by

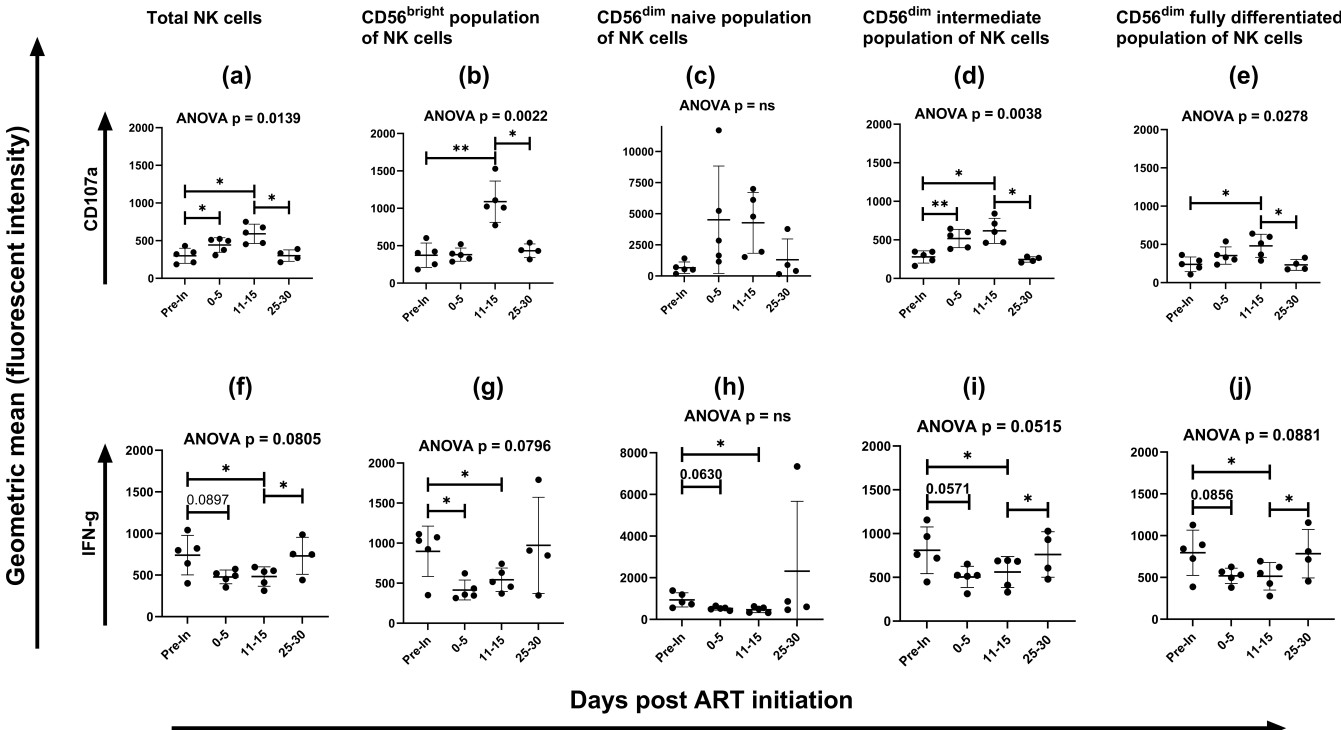

**FIG 2** Changes in NK cell effector functions (degranulation as defined by expression of CD107a and production of IFN-ɣ) from pre-infection time point to 25–30 days post ART initiation. Expression levels of CD107a (a through e) and IFN-ɣ (f through j) in total NK cell population (a and f) and in the different NK cell sub populations; (b and g) CD56^bright NK cell population, (c and h) CD56^dim naïve NK cells, (d and i) CD56^dim intermediate NK cells, and (e and j) CD56^dim fully differentiated NK cells across the different sample collection time points. Pre-In = sample collected before the first HIV-1 sero-positive test (pre-infection time point). Statistical test used between time points: two-tailed paired t test. Statistical test used across time points: unmatched, non-parametric one-way ANOVA with multiple comparisons [mean of each column with the mean of a control column (pre-infection time point); Geisser-Greenhouse correction used]. Each symbol indicates a participant. Line indicates mean value. Error bars indicate standard deviation. *$P < 0.05$, **$P$ 0.01.

the decrease in the expression of Siglec7 receptor on the NK cells. These observations point to an important role of Siglec7 receptor in NK cell function and further elucidation on the mechanism by which HIV-1 interferes with its expression could guide to a therapeutic target. The cytolytic function of NK cells is mediated by granzyme and perforin degranulation, leading to infected cells' apoptosis. Our results showed an increase in the degranulation of NK cells very early in HIV-1 infection. This was restored to the pre-infection levels between 15 and 30 days after ART initiation, suggesting ART-mediated suppression viral replication hence reduced NK cell degranulation. Our observations are consistent with a study that evaluated phenotypes and functionality of NK cells in ART-treated chronic HIV-1 infected individuals, in which expression of CD107a and production of granzyme B were significantly lower in the ART-treated HIV-1 infected individuals compared to the uninfected controls (9). In contrast to degranulation, we observed a significant decrease in the production of IFN-ɣ, a very potent anti-viral cytokine. IFN-ɣ is critical in controlling viral replication and dissemination. HIV-1 interference with NK cell production of this cytokine could be among the many ways of evading the immunity.

Our findings suggest that early initiation of ART is effective and may be important in boosting innate immunity and restoring some perturbations of NK cells resulting from HIV-1 infection. To elucidate the nuanced interactions and counteractions between NK cells and HIV-1, future investigations should aim to unveil the interplay between viral mechanisms and individual NK cell receptors and their associated signaling pathways. Understanding the persistence and restoration of specific NK cell markers and receptors post-ART can offer insights into potential therapeutic targets and strategies to bolster the innate immune response against HIV-1 and other viral infections.

## ACKNOWLEDGMENTS

We thank International AIDS Vaccine Initiative (IAVI) for funding the project (Investigator Initiated Research award given to M.M.) and HIV Pathogenesis Programme, University of KwaZulu-Natal for training in NK cell flow cytometric analysis. We also acknowledge the volunteers who participated in this study and the KAVI Institute of Clinical Research staff for their contribution to the study in various capacities. We also wish to acknowledge the training and support from "UANDISHI-Building Capacity for Writing Scientific Manuscripts" program at the Faculty of Health Sciences, University of Nairobi. This work was partially funded by IAVI with the generous support of United States Agency for International Development (USAID) and other donors; a full list of IAVI's major donors is available at iavi.org. The contents of this manuscript are the responsibility of the authors and do not necessarily reflect the views of USAID or the U.S. Government.

This study was funded by IAVI. The funder had no role in the study design, data collection and analysis, decision to publish, or preparation of the manuscript. The funder declared that no conflict of interest exists.

M.A. assisted with the study design conceptualization, performed all the laboratory bench work, data acquisition, data analysis, prepared the draft manuscript for review by the co-authors and prepared the final manuscript for submission. M.M. designed the study, applied and obtained funding from IAVI, assisted in the training and optimization of the laboratory procedures used in the study. C.T. and K.N. assisted with training of laboratory procedures and optimization, reviewed the draft manuscripts. D.M., R.L., G.O., G.M., D.O., K.C. and O.A. helped in the conceptualization and design of the study. All authors reviewed the draft manuscripts and approved the final version for submission.

This work was presented as an oral presentation on the 24th of January, 2023 at the UoM/UoN STD/HIV/SRH COLLABORATIVE RESEARCH GROUP ANNUAL REVIEW MEETING 2023 held in Nairobi, Kenya. The work was also presented as a poster at the 18th International Congress of Immunology held from 27th November to 2nd December 2023 in Cape Town, South Africa

## AUTHOR AFFILIATIONS

[1]KAVI Institute of Clinical Research, University of Nairobi, Nairobi, Kenya

[2]Department of Medical Microbiology and Immunology, University of Nairobi, Nairobi, Kenya

[3]HIV Pathogenesis Programme, University of KwaZulu-Natal, KwaZulu-Natal, South Africa

[4]Division of Surgical Outcomes and Precision Medicine Research, Department of Surgery, University of Minnesota Twin Cities, Twin Cities, Minnesota, USA

[5]International AIDS Vaccine Initiative, New York, New York, USA

[6]Faculty of Health Sciences, School of Pathology, University of Witwatersrand, Witwatersrand, South Africa

[7]Centre for HIV and STIs, National Institute for Communicable Diseases, Johannesburg, South Africa

## AUTHOR ORCIDs

Matrona Akiso  http://orcid.org/0000-0002-3665-9304
Marianne W. Mureithi  http://orcid.org/0000-0001-9119-3167

## FUNDING

| Funder | Grant(s) | Author(s) |
| --- | --- | --- |
| International AIDS Vaccine Initiative (IAVI) | | Marianne W. Mureithi |

## AUTHOR CONTRIBUTIONS

Matrona Akiso, Conceptualization, Data curation, Formal analysis, Investigation, Methodology, Project administration, Validation, Visualization, Writing – original draft, Writing – review and editing | Daniel Muema, Formal analysis, Writing – review and editing | Robert Langat, Conceptualization, Formal analysis, Supervision, Visualization, Writing – review and editing | Kewreshini K. Naidoo, Conceptualization, Methodology, Writing – review and editing | Geoffrey Oino, Conceptualization, Project administration, Writing – review and editing | Gaudensia Mutua, Conceptualization, Writing – review and editing | Christina Thobakgale, Conceptualization, Formal analysis, Methodology, Writing – review and editing | Daniel Ochiel, Conceptualization, Writing – review and editing | Kundai Chinyenze, Conceptualization, Writing – review and editing | Omu Anzala, Conceptualization, Supervision, Writing – review and editing | Marianne W. Mureithi, Conceptualization, Data curation, Formal analysis, Funding acquisition, Methodology, Project administration, Resources, Supervision, Validation, Visualization, Writing – review and editing

## DATA AVAILABILITY

All relevant data generated in this study are available within the paper and its supporting information files at KAVI Institute of Clinical Research and can be obtained from the corresponding author on reasonable request.

## ETHICS APPROVAL

This study was approved by KNH/UoN/ERC (P61/02/2017). All study participants signed an informed consent form prior to enrollment to participate in the study.

## ADDITIONAL FILES

The following material is available online.

### Supplemental Material

**Fig. S1 (Spectrum03570-23-s0001.tif).** Occurrences and changes in NK cell frequencies, CD4 cell counts, and HIV viral loads across the different study time points.
**Fig. S2 (Spectrum03570-23-s0002.tiff).** Gating strategy for defining the different NK cell subsets.
**Fig. S3 (Spectrum03570-23-s0003.tif).** Frequency of total NK cells expressing phenotypic surface markers.
**Fig. S4 (Spectrum03570-23-s0004.tif).** Changes in the frequencies of functional NK cells at the different time points.
**Fig. S5 (Spectrum03570-23-s0005.tiff).** Flow jo analysis and gating for participant NK02001 for the NK cell functional panel.
**Legends (Spectrum03570-23-s0006.docx).** Titles for the supplemental figures and tables
**Additional experimental details (Spectrum03570-23-s0007.docx).** Detailed materials and methodology used in the study.
**Table S1 (Spectrum03570-23-s0008.pdf).** Demographic data of the study participants.
**Table S2 (Spectrum03570-23-s0009.pdf).** Anti-human antibodies used for flow cytometry staining.

### Open Peer Review

**PEER REVIEW HISTORY (review-history.pdf).** An accounting of the reviewer comments and feedback.

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
