## [Reviewer comments · Microbiology Spectrum]

Microbiology Spectrum

Early Antiretroviral Therapy and its Impact on Natural Killer Cell Dynamics in HIV-1 infected Men who have sex with men: A cross-sectional pilot study evaluating the impact of early ART initiation on NK cell perturbation in HIV infection

Matrona Akiso, Robert Langat, Daniel Muema, Kewreshini Naidoo, Geoffrey Oino, Gaudensia Mutua, Christina Thobakgale-Tshabalala, Daniel Ochiel, Kundai Chinyenze, Omu Anzala, and Marianne Mureithi

Corresponding Author(s): Marianne Mureithi, University of Nairobi

Review Timeline:

Submission Date:	October 9, 2023
Editorial Decision:	November 12, 2023
Revision Received:	January 15, 2024
Accepted:	January 17, 2024

Editor: Takamasa Ueno

Reviewer(s): The reviewers have opted to remain anonymous.

Transaction Report:

DOI: <https://doi.org/10.1128/spectrum.03570-23>

Re: Spectrum03570-23 (Early Antiretroviral Therapy and its Impact on Natural Killer Cell Dynamics in HIV-1 infected Men who have sex with men: A cross-sectional pilot study evaluating the impact of early ART initiation on NK cell perturbation in HIV infection)

Dear Dr. Marianne Mureithi:

One weakness of this study is small sample size (5 males) involving merely flow cytometric analyses, while there are some interesting observations presented. This limitation was not improved in the revised manuscript although other points seemed to be adequately addressed. Considering the limited scope of this study, the category "Observations", instead of "Research Article", should be suitable. If the authors want revise the manuscript, please reformat it to "Observations".

Revision Guidelines

Sincerely,
Takamasa Ueno
Editor
Microbiology Spectrum

Re: Spectrum03570-23R1 (Early Antiretroviral Therapy and its Impact on Natural Killer Cell Dynamics in HIV-1 infected Men who have sex with men: A cross-sectional pilot study evaluating the impact of early ART initiation on NK cell perturbation in HIV infection)

Dear Dr. Marianne Mureithi:

Your manuscript has been accepted, and I am forwarding it to the ASM production staff for publication. Your paper will first be checked to make sure all elements meet the technical requirements. ASM staff will contact you if anything needs to be revised before copyediting and production can begin. Otherwise, you will be notified when your proofs are ready to be viewed.

Sincerely,
Takamasa Ueno
Editor
Microbiology Spectrum